# Graphene Nanoplatelets Hybrid Flame Retardant Containing Ionic Liquid and Ammonium Polyphosphate for Modified Bismaleimide Resin: Excellent Flame Retardancy, Thermal Stability, Water Resistance and Unique Dielectric Properties

**DOI:** 10.3390/ma14216406

**Published:** 2021-10-26

**Authors:** Yan Wang, Xining Jia, Hui Shi, Jianwei Hao, Hongqiang Qu, Jingyu Wang

**Affiliations:** 1National Engineering Technology Research Center of Flame Retardant Materials, School of Materials Science and Engineering, Beijing Institute of Technology, Beijing 100081, China; WangYan09302021@163.com (Y.W.); xiningjia97@126.com (X.J.); 3220191073@bit.edu.cn (H.S.); 2The Flame Retardant Material and Processing Technology Engineering Research Center of Hebei Province, College of Chemistry and Environmental Science, Hebei University, Baoding 071002, China; 3School of Materials Science and Mechanical Engineering, Beijing Technology and Business University, Fucheng Road 11, Beijing 100048, China

**Keywords:** graphene nanoplatelets, ionic liquid, ammonium polyphosphate, bismaleimide resin, flame retardancy, dielectric properties

## Abstract

To achieve the requirements of modified bismaleimide resin composites in electronic industry and high energy storage devices, flame retardancy, water resistance and dielectric properties must be improved. Hence, a highly efficient multifunctional graphene nanoplatelets hybrid flame retardant is prepared by ionic liquid graphite and ammonium polyphosphate. The preparation processes of the flame retardants are simple, low energy consumption and follow the green chemical concept of 100% utilization of raw materials, compared with chemical stripping. The bismaleimide resin containing 10 wt.% of the flame retardant show good flame retardancy, resulting in the limiting oxygen index increases to above 43%, and the peak heat release rate, total heat release and total smoke release decrease by 41.8%, 47.8% and 52.3%, respectively. After soaking, mass loss percentage of the modified bismaleimide resin only decreases by 0.96%, the dielectric constant of the composite increases by 39.4%, and the dielectric loss decreases with the increase of frequency. The hybrid flame retardants show multifunctional effect in the modified bismaleimide resin, due to the physical barrier, the chemical char-formation, hydrophobicity and strong conductivity attributed to co-work of Graphene nanoplatelets, ammonium polyphosphate and ionic liquid.

## 1. Introduction

With the rapid development of electronic industry, high dielectric constant materials with excellent performance of blocking carriers, storing electric energy and uniform electric field, stealth/wave transmission integration, etc., have been widely used in embedded capacitors, photoelectric/electro-optic devices, microwave devices and electromagnetic interference shielding [1,2]. Compared with traditional inorganic materials (such as ferroelectric ceramics) with high dielectric constant, and polymers with low dielectric constant, polymer-based nanocomposites have the advantages of good dielectric properties, excellent mechanical properties, simple molding process and light weight [3]. 

Bismaleimide resin (BMI) has been widely used in microelectronic industry and aerospace field due to its excellent heat resistance, good processing characteristics and high mechanical strength. Since BMI is very brittle, it is usually toughened. However, the addition of toughening agent often makes it release more heat and smoke during combustion [4]. To enable BMI to meet the special application requirements in the above areas, it is necessary to modify it, such as optimizing its flame retardancy and dielectric properties [5,6]. At present, the research on the composite properties based on flame retardant BMI mainly focuses on the improvement of mechanical properties and thermal conductivity. In addition, many efforts have been proposed, such as hierarchical MoS_2_@TiO_2_ [7], polyphosphazene modified by g-C_3_N_4_ tubes [4], diallyloxydiphenyldisulfide and three-dimensional porous boron nitride framework [8,9] had been prepared to improve the flame retardant, thermal conductivity and mechanical properties of BMI. Unfortunately, these research rarely involve the optimization of dielectric properties, especially there are few studies of resin-based composites with the flame retardant and high dielectric constant, which is required in the field of microelectronics [10]. Therefore, combining nano conductivity filler with traditional flame retardants should be one of the effective ways to achieve this goal.

Graphene nanoplatelets (GNPs) have the potential to endow the high dielectric properties for polymer composites owing to their excellent conductivity. The few-layer GNPs can be obtained through ionic liquids (ILs) assisting mechanical exfoliation (such as ball milling) of raw graphite or expanded graphite in liquid-phase, based on matching to the surface tension of imidazolium IL and graphene (about 40 mJ m^−2^) [11]. Hence, this preparation method is simple, low-cost and higher exfoliation efficiency compared with the traditional 1-Methyl-2-pyrrolidinone (NMP) exfoliating graphite to GNPs [12]. GNPs obtained by this way processes fewer defects and damage to the electronic structure of graphene than those prepared by Hummer’s method [11,13]. In addition, ILs generally exhibit high thermal stability (extending to temperatures above 300 °C) [14] and contain phosphorus and nitrogen, such as 1-alkyl-3-methylimidazolium hexafluoro phosphates, are expected to act as additives to endow the synergy action with traditional flame retardants used in polymer composites. For example, an IL containing phosphorus with ammonium polyphosphate (APP) of (IL:APP = 1:5 or 1:1) addition amounts can endow polypropylene or poly (lactic acid) with fire safety [15,16]. Meanwhile, the incorporation of IL/APP have improved mechanical properties of composites. Based on the above results, ILs with suitable structure may show synergism with APP, and is also expected to exhibit more other positive effects for modified BMI in the presence of GNPs.

Some studies have proved that the flame retardants containing phosphorus compounded with GNPs should play the role of chemical catalytic carbonization and physical barrier [17]. Ammonium polyphosphate (APP) is an environmentally friendly traditional flame retardant additives because of its high phosphorus content, low cost, low toxicity and low corrosion. The production and consumption of APP in China have rapidly increased in 2019 to 10,800 tons with the annual growth of 19.6% for recent four years. Its high phosphorus content (about 32%) makes it have high chemical catalytic char forming ability by the interaction between degraded polyphosphoric acid and fragments containing oxygen from polymer substrate through esterification and cross-linking reactions. However, its slightly solubility in water makes it easy to migrate out of the substrate [18,19]. Many water-resistant studies of APP indicate that nano SiO_2_ [20], KH550 and silicon resin [21] could be coated on the surface of APP, tetraethoxysilane and octyltriethoxysilane as precursors hydrolyze and condense to form a dense polysiloxane layer structure on the surface of APP to realize its hydrophobicity [22]. The above results indicate that it may be sophisticated solution to encapsulate APP with hydrophobic material, which usually leads to cost up of modified APP. Therefore, the IL-APP, formed by the reaction of imidazolium cations of IL binding with anions of APP in the liquid toughening agent of BMI in the presence of GNPs, is expected to improve the water resistance and flame retardant efficiency, and make use of a large number of micro capacitors formed between the GNPs to construct a super capacitive network, thus increasing the dielectric constant of BMI composites [23]. 

In this paper, we focus on the major problem of preparing graphene nanoplatelets hybrid flame retardant GNPs/IL-APP from graphite or expanded graphite using IL assisted mechanical exfoliation and binding APP in the liquid toughening agent 4,4′- (propane-2,2-diyl) bis (2-allylphenol) (DABA) during the preparation processes of BMI composites. Based on characterization of the structure, morphology and components of GNPs/IL-APP, the effect of this hybrid flame retardant on the thermal stability, flame retardancy, water resistance and dielectric properties of BMI composites are studied.

## 2. Experimental Section

### 2.1. Materials

Powder graphite (MPG, ADT 005, 90~99.5% Particle size distribution: D10 1.5~2.5 μm; D50 4~6 μm; D90 8~11 μm) and expandable graphite (EG, ADT 249, >95%, 50 mesh) were obtained from Shijiazhuang ADT Carbonic Material Factory (Hebei, China). Ionic liquid 1-butyl-3-methylimidazolium hexafluoro phosphate (IL, [Bmim]PF_6_, ≥97%) was purchased from Sigma-Aldrich Co., Ltd. (St. Louis, MI, USA). Ammonium polyphosphate (APP) (Type II) was obtained from JLS Flame Retardants Chemical Co., Ltd. (Hangzhou, China). 4,4′-Bismaleimidodiphenylmethane (BMI, 98%) and 4,4′-(Propane-2,2-diyl) bis(2-allylphenol) (DABA, 90%) were provided by Meryer Chemical Technology Co., Ltd. (Shanghai, China). 

### 2.2. Preparation of MPGNPs and EDGNPs

Firstly, expanded graphite (EDG) was prepared by heating 5 g of expandable graphite (EG) to 800 °C in a tubular furnace in air atmosphere for 1 min. 1 g of MPG and 1 g of EDG were added into two 500 mL zirconia ball milling tanks, respectively, then 100 mL of deionized water suspension containing 1 g of IL was added, respectively, and 5, 15 and 25 zirconia ball milling beads with diameters of 15 mm, 10 mm and 5 mm were added, respectively. The ball milling was carried out at the speed of 300 r min^−1^ for 2 h. After that, the substances in the ball milling tanks (Miqi Equipment Co., Ltd., Changsha, China) were completely transferred to different beakers for 4 h ultrasonic cleaning, the products were dried to obtain MPGNPs and EDGNPs containing GNPs bound with IL, respectively. 

The suspension after ultrasonic cleaning was centrifuged at 8000 r min^−1^ for 15 min, then the supernatant was filtered and washed with deionized water on polytetrafluoroethylene quantitative filter paper. After repeat washing, the filter paper was dried at 75 °C until the quality did not decline. The yields of GNPs from MPG and EDG were 2.26% and 2.72%, respectively.

### 2.3. Preparation of DBMI Composite and GNPs Hybrid APP-IL

The DBMI composites containing GNPs/IL-APP hybrid flame retardant and control sample were prepared according to the ratios shown in Table 1. The toughener DABA was heated to 130 °C in a beaker to obtain good fluidity. The chemical formula of DBMI’s curing reaction is shown in the Figure 1. All additives were added slowly, stirred at 90 r/min for 30min to disperse evenly, and then the temperature was gradually increased to 150 °C. A small amount of BMI was added for many times. The pre-polymerization was carried out until the system was translucent. After that, product was put into a vacuum oven(DZF-6020, GONGYI YUHUA Instrument Co., Ltd., Henan, China) preheated at 150 °C for 45 min, then it was poured into a mold and cured by a curing process of 150 °C/1 h–170 °C/1 h–190 °C/2 h–210 °C/2 h. After curing, de-molding and post-treatment at 240 °C for 40 min. From this, neat DBMI as control and DBMI composites were obtained, respectively [24].

It is worth noting that when IL and APP were heated to 130 °C in DABA, the ion exchange reaction (discussed later) occurred between imidazolium cations of IL and anions of APP, which made IL tightly bound on the surface of APP. This binding structure was named IL-APP, and the van der Waals force between IL and GNPs made GNPs indirectly surround APP molecular chain or crystal, Finally, a new flame retardant system of GNPs hybrid APP-IL was formed. 

### 2.4. Measurements and Characterization

Atomic force microscopy (AFM, Dimension FastScan Bio, Bruker Ltd., Karlsruhe, Germany) was used to characterize the micro scale of GNPs, such as thickness, number of layers, maximum radius, aspect ratio and so on. 

Raman spectroscopy (RM 2000, Renishaw Ltd., Gloucestershire, UK) was used to characterize the micro states of GNPs such as defect density, using 633 nm laser excitation.

Nuclear Magnetic Resonance (NMR, Advance2B, Bruker, Germany) was used to test the chemical shifts of the specified elements to determine the chemical environment of the elements. In this paper, the possible chemical reactions between IL and APP were characterized by ^31^P NMR and ^1^H NMR using 256 scans, 40 s pulse time, 20 s pulse delay and 90° pulse angle.

Scanning electron microscope analysis (SEM, Hitachi S-4800, Hitachi Ltd., Tokyo, Japan) was used to obtain information on the surface morphology of the hybrid flame retardant and the residual char of DBMI composites.

Using X-ray Energy Dispersive Spectroscopy (XEDS, Hitachi S-4800, Hitachi Ltd., Tokyo, Japan) on the basis of SEM, the composition and content of the hybrid flame retardant were characterized using Si (Li) detector, the energy resolution is better than 133 eV, and the detection range is Be(Z = 4)~U(Z = 92).

Thermogravimetric Analysis (TGA, TG 209 F1 Iris^®^, NETZSCH, Germany) was used to test the thermal stability of DBMI at a heating rate of 10 °C/min from 50 to 700 °C in N_2_ atmosphere.

The Limiting oxygen index (LOI) test shows the minimum oxygen concentration required for continuous combustion of the material in a mixed atmosphere of nitrogen and oxygen. It is usually used to evaluate the flammability of materials. In this study, LOI were measured by an FTA-II instrument (Rheometric Scientific Ltd., Reichelsheim, Germany) with specimen dimensions of 130 × 6.5 ×3 mm^3^, according to ASTM D 2863-08. 

The cone calorimeter’s (CONE) test environment is similar to the real combustion environment and has a good correlation with the combustion behavior of materials in large-scale combustion test or real combustion. In this study, CONE (FTT, fire testing technology apparatus) measurement was performed under 50 kW/m^2^ external radiant heat flux conforming to ISO 5660 protocol. The specimen dimension is 100 × 100 × 3 mm^3^.

In order to determine the water resistance of the DBMI composites, the specimens were put into distilled water at 60 °C and were kept for 120 h, and the water was replaced every 24 h. The treated specimens were subsequently taken out and dried them to keep the mass unchanged in a vacuum oven(DZF-6020, GONGYI YUHUA Instrument Co., Ltd., Henan, China), and the mass loss percentages were measured. The water resistance of the DBMI composites was evaluated by the change of mass loss percentages and LOI values.

The impedance analyzer (Wayne Kerr 6500B, Wayne Kerr Electronics, West Sussex, UK) was used to test the dielectric properties of DBMI composites to obtain the real part of the relative complex permittivity (ε_r_’) and tangent value of dielectric loss angle (tan δ) diagram line of relationship with AC frequency. The test frequency is 10^3^–10^6^ Hz, and the samples are all discs with diameter of 50 mm and thickness of 3.5–4.5 mm.

## 3. Results and Discussion

### 3.1. Morphology and Chemical Compositions of GNPs Hybrid Flame Retardants

As shown in Figure 2 and Table 2, AFM was used to characterize the micro scale of GNPs. In the selected region, the thickness of GNPs in MPGNPs was about 7.6 nm and the surface radius was about 0.1 μm. The aspect ratio was about 26.4, the thickness of GNPs in EDGNPs was 2.7 nm, and surface radius was about 0.1 μ m. The aspect ratio was about 98.4, and the surface roughness of both GNPs was obvious. 

Raman spectroscopy is an important diagnostic tool for the analysis of graphite-like materials such as graphene. Raman spectra of MPGNPs and EDGNPs are shown in Figure 2e,f. Taking MPGNPs as an example, the G peak of MPGNPs at 1581.09 cm^−1^ is caused by the in-plane vibration of sp^2^ carbon atom, which is the characteristic peak of graphene [25]. The G’ peak of 2685.40 cm^−1^ is weaker than that of G, which indicates that the GNPs are mainly multilayer graphene. The spectrum of EDGNPs is similar. At the same time, the intensities of D peak and D’ peak at about 1329.52 cm^−1^ and 1619.74 cm^−1^ are lower, I(D)/I(G) of MPGNPs is 0.35, and I(D)/I(G) of EDGNPs is 0.18, which indicates that EDGNPs have fewer defects than MPGNPs [26,27]. 

The reaction between IL and APP and the preparation process of GNPs/IL-APP hybrid flame retardant are shown in Figure 3a,b, respectively. In order to reveal the ion exchange bind reaction between imidazolium cations of IL and anions of APP in DABA, IL and APP were added to DABA at 130 °C and stirred (90 r/min) for 30 min. The uniformly dispersed liquid was dissolved in DMSO-*d*_6_ for ^31^P NMR and ^1^H NMR characterization. The test results are as follows: APP ^31^P NMR (162 MHz, Deuterium Oxide) δ = 2.00, −7.85, −22.96. IL ^31^P NMR (162 MHz, DMSO-*d*_6_) δ = −131.01, −135.36, −139.80, −144.19, −148.57, −152.96, −157.36. IL-APP ^31^P NMR (162 MHz, DMSO-*d*_6_) δ = −1.08, −2.92, −8.47, −15.87, −131.34, −135.72, −140.11, −144.50, −148.90, −154.32, −157.67. The ^31^P NMR spectrum of IL-APP showed that new peaks were generated in the region corresponding to APP and IL, and the peak area changed. The obvious changes of the relative strength of the new peaks and the peaks around APP indicate that new chemical bonds between IL and APP have been produced, which makes the chemical environment of P and H elements change [28]. In order to show the assembly of APP by IL and GNPs more intuitively, the process has been repeated in a system with water of 75 °C as dispersant. Taking the assembly of APP by MPGNPs and IL as an example, the SEM and XEDS were characterized by removing water. The selected area in Figure 3c is scanned with C element surface, and the mapping layer is covered on the SEM picture of the area. The dense area of element C should be graphite containing GNPs adsorbed and the sparse area should be IL bound on APP crystal. It can be seen that IL bound on APP is uniform, and the van der Waals force between GNPs and IL makes GNPs more closely distributed around the APP crystal, forming a micron dispersion structure.

### 3.2. Thermal Stability of the Flame Retardant Composites

Compared with Neat DBMI, DBMI composites with GNPs/IL-APP has higher thermal stability. TGA and DTG curves under nitrogen atmosphere are shown in Figure 4, and the detailed data are summarized in Table 3. The thermal decomposition temperature of loss weight of 5 wt.% (T_d,5%_) for flame retardant DBMI composites decrease from 409 °C for Neat DBMI to 370~376 °C, the maximum mass loss temperature (T_max_) also decreases with addition of hybrid flame retardants. This is due to that APP containing flame retardants can accelerate the thermal degradation and char-forming of DBMI matrix at a relative lower temperature [29]. At 450 °C, there are little difference in the residual mass of composites samples (R_450_
_°C_), but after that the residue char at 700 °C (R_700_
_°C_) reaches about 60% originated from the catalytic carbonization of GNPs hybrid flame retardants. It means that organic-inorganic hybrid flame retardants can play double barrier effects [17], which endows DBMI with good thermal stability at high temperature. Compared with DBMI/MPG/IL-APP, WLR_max_ of the other composites is slightly reduced, and their R_700_
_°C_ significantly increase, which mean that EDG as graphite intercalated compounds and GNPs with larger specific surface area have better char forming ability for composites than that of MPG [30,31], and the increase of char residue leads to the enhanced barrier effect, further decomposition of the composite is avoided [32]. 

### 3.3. Flame Retardant Properties of DBMI Composites

The addition of GNPs hybrid flame retardant can improve the flame retardancy of DBMI. The LOI and CONE test results are shown in Figure 5 and Table 4. The test results are the average of five measurements, and the systematic error generated by the experimental equipment is less than **±** 0.2%. LOI data in Figure 5a indicate that DBMI composites with GNPs hybrid flame retardant have the highest LOI of 43.7% and 43.8%, respectively, which are much higher than that of Neat DBMI (~%). This indicates that the flame retardant containing GNPs exfoliated from MPG and EDG has higher flame retardant efficiency. Furthermore, compared with DBMI/APP and DBMI/IL-APP samples at similar loading, GNPs and APP have synergistic flame retardant effect in DBMI [33,34]. 

CONE test results in Figure 5b–d and Table 3 show that DBMI composites containing GNPs hybrid flame retardant release less heat and smoke, and don’t have obvious effect on time to ignition (TTI) during combustion. Compared with Neat DBMI and the samples containing without exfoliated MPG and EDG, the peak heat release rate (PHRR) of DBMI/MPGNPs/IL-APP is decreased to 184.38 kW·m^−2^ from 327.10 kW·m^−2^ of Neat DBMI by 43.6%, and the PHRR of DBMI/EDGNPs/IL-APP also decreases significantly. The total heat release (THR) and total smoke release (TSR) of DBMI/EDGNPs/IL-APP are reduced the most, reaching 47.8% and 52.3%, respectively, which are consistent with the change of average mass loss rate (av-MLR). For average effective combustion heat (av-EHC), no significant difference can be found among composites samples, which indicates that the flame retardants mainly play the role of condensed phase through higher catalytic carbonization efficiency, thus significantly reducing the combustion intensity [35]. 

As shown in Figure 6, the carbon residue of DBMI/MPGNPs/IL-APP was denser and produced fewer cracks than that of Neat DBMI and DBMI/MPG/IL-APP. From the micro morphology of char residue in Figure 7, it can be seen that although the main part of char residue in Neat DBMI is dense, there are large pores (>200 μm), which may be the reason of its higher mass loss rate and poor flame impact resistance. The char residue of DBMI/MPGNPs/IL-APP and DBMI/EDGNPs/IL-APP present more compact and multi-layer structure than that of the other composites to play the best role of barrier, which strongly support the test results of combustion behaviors [36,37]. 

### 3.4. Water Resistance of the Flame Retardant Composites

GNPs and IL can mutually prevent APP from migrating from DBMI substrate in water environment. The relationship between mass lost and soaking time for the DBMI composites with GNPSs hybrid flame retardant and control samples (DBMI/APP and DBMI/IL-APP) is shown in Figure 8a. The results of LOI change and mass lost percentage of DBMI composites after water resistance tests are listed in Figure 8b and related data are summarized in Table 5. It is clear that the mass loses of DBMI/APP and DBMI/IL-APP is up to 2.36 wt.% and 2.09 wt.%, respectively, after water resistance test. When introducing MPG or EDG to system, no significant change of water resistance for DBMI/MPG/IL-APP and DBMI/EDG/IL-APP is found, although the latter decrease slightly. By contrast, the mass loss of DBMI/MPGNPs/IL-APP and DBMI/EDGNPs/IL-APP is markedly reduced at each time period, which indicates that GNPs exfoliated can endow APP better water resistance and lower water migration rate under the cooperation of IL. Namely, it is the co-work function between the barrier effect of few-layer GNPs and hydrophobic IL. 

Therefore, from the LOI test results of water emigrated samples it can be found that DBMI/MPGNPs/IL-APP and DBMI/EDGNPs/IL-APP have relatively higher LOI. While the LOI of DBMI/MPG/IL-APP and DBMI/EDG/IL-APP are significantly decreased, which is consistent with the results of mass lost and non-soaked samples. In particular, the synergistic water and flame resistance between EDGNPs and IL for APP in DBMI composites is better than that of MPGNPs and IL, because EDGNPs has smaller micro scale and more GNPs. 

### 3.5. Dielectric Properties of Flame Retardant Composites

GNPs can endow DBMI with higher dielectric constant. Figure 9a, shows the relationships between the dielectric constant (ε_r_’) and frequency of each sample. In 10^3^~10^6^ Hz AC frequency range, the ε_r_’ of samples containing graphite all increased compared with that of Neat DBMI. In particular, the ε_r_’ of DBMI/EDG/IL-APP and DBMI/EDGNPs/IL-APP increases the most, about 3.34 and 3.50 at 10^6^ Hz, which increases by 33.1% and 39.4%, respectively, then 2.51 of Neat DBMI. This is due to that the dielectric properties of the nanocomposites follow the percolation mode [38]. Since EDGNPs are exfoliated from EDG (expanded by EG) and have higher exfoliation effective than MPGNPs, Compared with MPG and MPGNPs, the aspect ratio of EDG and EDGNPs was larger, which formed more graphene-polymer interface and more graphene-graphene micro capacitance. It is reason for more dielectric constant increase [23]. 

The relationships between dielectric loss tangent (tanδ) and frequency are shown in the Figure 9b, which is different from that of GNPs in epoxy resin [39]. Within tested frequency, tan δ of samples containing graphite were higher than Neat DBMI, tanδ of DBMI/EDG/IL-APP, DBMI/MPGNPs/IL-APP, DBMI/EDGNPs/IL-APP decreased with the increase of frequency. DBMI/MPGNPs/IL-APP and DBMI/EDGNPs/IL-APP decreased the most, from 0.045 and 0.050 at 10^3^ Hz to 0.019 and 0.018 at 10^6^ Hz, respectively. However, DBMI/MPG/IL-APP showed a trend of first increase and then decrease. This may be due to larger additive size, which reduce the relaxation polarization loss [40]. According to the characterization of the micro morphology of GNPs, GNPs with lager aspect ratio increased the conductivity loss caused by the high leakage current formed by the direct connection between high conductive materials [41]. For DBMI/MPG/IL-APP, the graphite without expansion or exfoliation was unevenly distributed in the substrate, there was less direct connection between the graphite, which leaded to lower conductivity loss. However, relaxation polarization loss is higher, which is further manifested as tan δ. In the low frequency region, the values are lower but show an upward trend, while in the high frequency region, the values show a downward trend but ultimately higher than those of other groups [1,41].

## 4. Conclusions

In summary, a highly efficient multifunctional graphene nanoplatelets hybrid flame retardant GNPs/IL-APP was prepared by IL mechanical exfoliating graphite or expanded graphite and ion exchange binding APP. The results of TGA show that the residual char of DBMI composite with 10 wt.% of EDGNPs/IL-APP reaches 59.9% at 700 °C. LOI and CONE test results show that GNPs (MPGNPs or EDGNPs) can produce synergistic flame retardant effect with IL-APP, greatly reduce the heat and smoke release. It can be contributed to the physical barrier of GNPs and the chemical char-formation of APP, namely condensed phase action mechanism. The GNPs/IL-APP can also exhibit excellent water resistance, high dielectric constant and lower dielectric loss for DBMI composites, due to hydrophobicity and strong conductivity attributed to co-work of few-layer GNPs and IL-APP. The preparation of the hybrid flame retardant in this paper has the advantages of simple process, 100% utilization of raw materials and low energy consumption. It can provide a new way of organic-inorganic composite for the multifunctional DBMI and expand the application scenarios of composites. 

## Figures and Tables

**Figure 1 materials-14-06406-f001:**
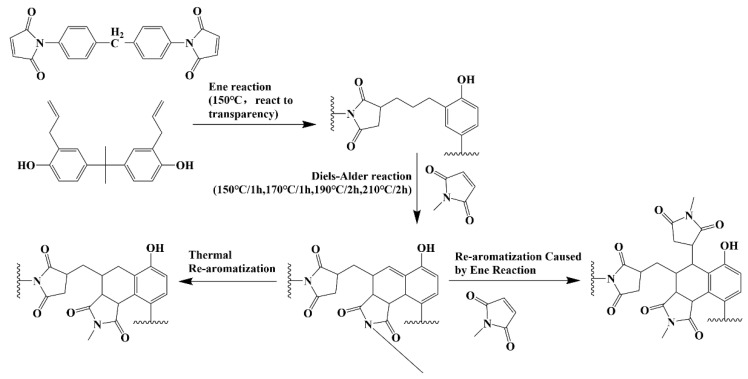
The chemical formula of DBMI’s curing reaction.

**Figure 2 materials-14-06406-f002:**
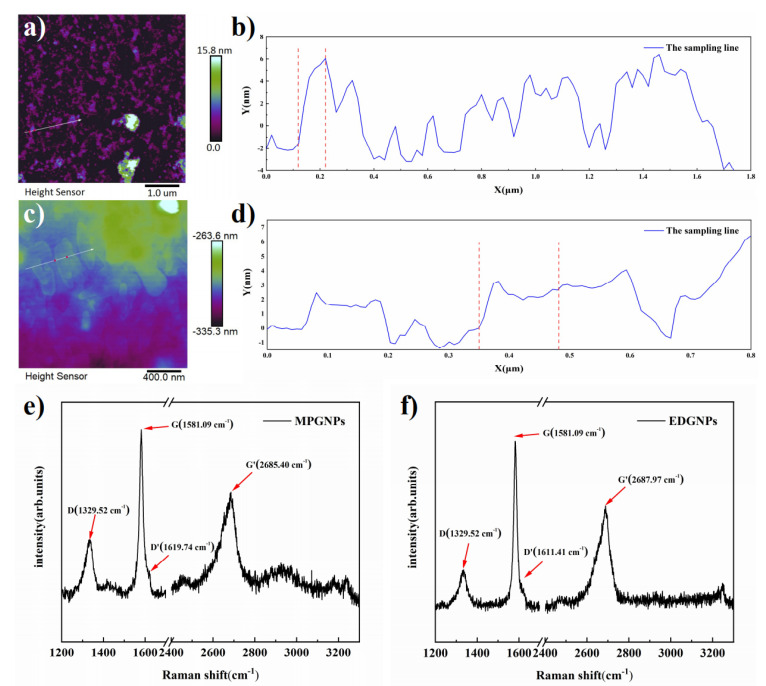
Typical tapping mode AFM image of MPGNPs (**a**) and EDGNPs (**c**), and the section analysis of MPGNPs (**b**) and EDGNPs (**d**); the Raman spectra test results of MPGNPs (**e**) and EDGNPs (**f**).

**Figure 3 materials-14-06406-f003:**
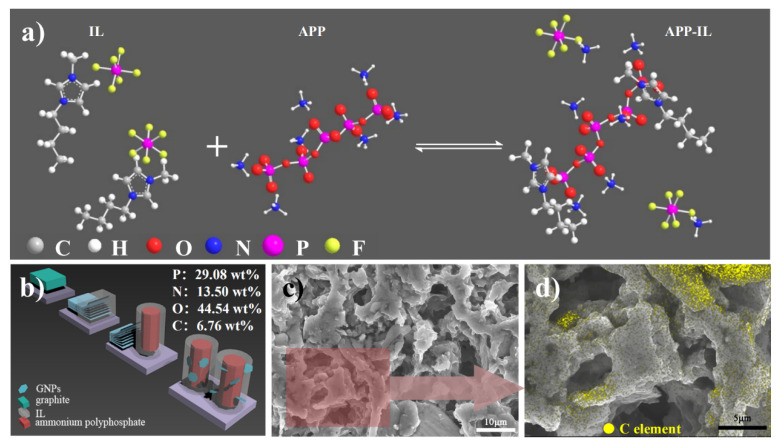
(**a**) Ion exchange reaction between IL and APP, (**b**) preparation process of PNGs/IL-APP hybrid flame retardant, (**c**) SEM image of MPGNPs and IL assembling APP, and (**d**) C element mapping of selected area.

**Figure 4 materials-14-06406-f004:**
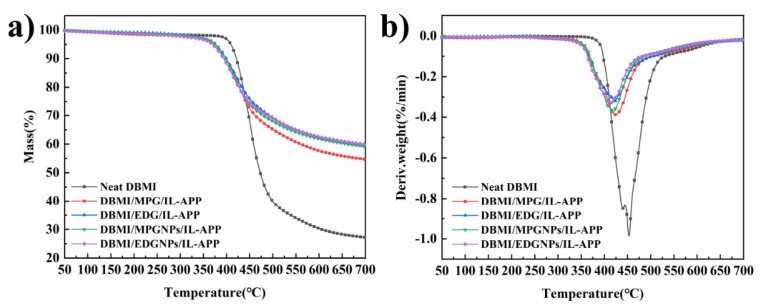
TGA curve (**a**) and DTG curve (**b**) of Neat DBMI, DBMI/MPG/IL-APP, DBMI/EDG/IL-APP, DBMI/MPGNPs/IL-APP and DBMI/EDGNPs/IL-APP.

**Figure 5 materials-14-06406-f005:**
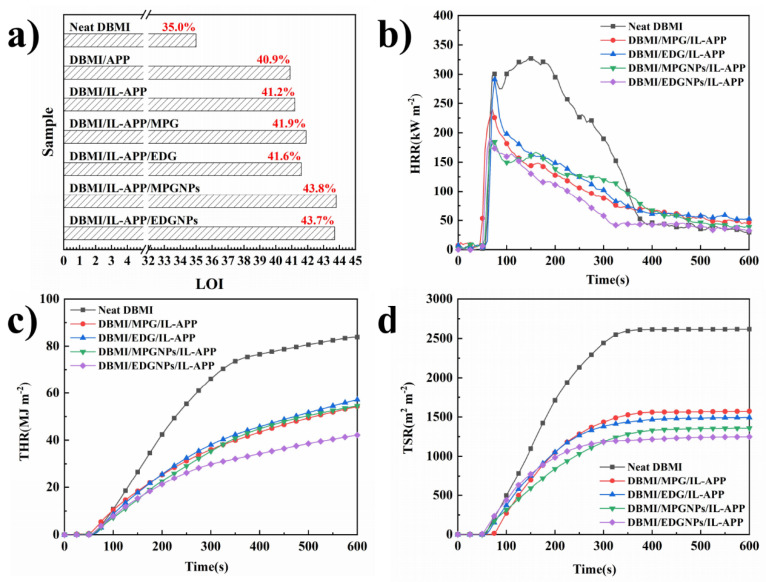
LOI (**a**) test results, HRR (**b**), THR (**c**) and TSR (**d**) curves of Neat DBMI, DBMI/MPG/IL-APP, DBMI/EDG/IL-APP, DBMI/MPGNPs/IL-APP, DBMI/EDGNPs/IL-APP in CONE test.

**Figure 6 materials-14-06406-f006:**
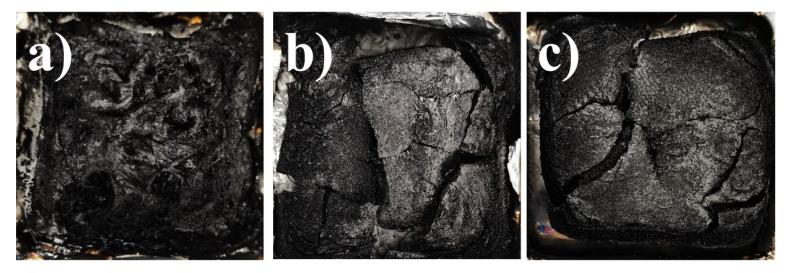
The digital photos of carbon residue of Neat DBMI (**a**), DBMI/MPG/IL-APP (**b**), DBMI/MPGNPs/IL-APP (**c**) were measured by CONE.

**Figure 7 materials-14-06406-f007:**
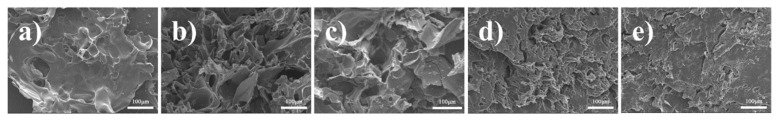
The SEM images of carbon residue of Neat DBMI (**a**), DBMI/MPG/IL-APP (**b**), DBMI/EDG/IL-APP (**c**), DBMI/MPGNPs/IL-APP (**d**), DBMI/EDGNPs/IL-APP (**e**) were measured by CONE.

**Figure 8 materials-14-06406-f008:**
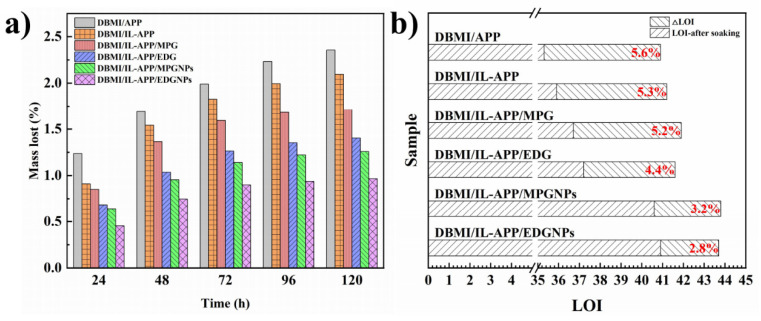
The relationships between mass lost and soaking time (**a**) and LOI changes before and after 120 h immersion at 60 °C (**b**) of DBMI composites and control samples.

**Figure 9 materials-14-06406-f009:**
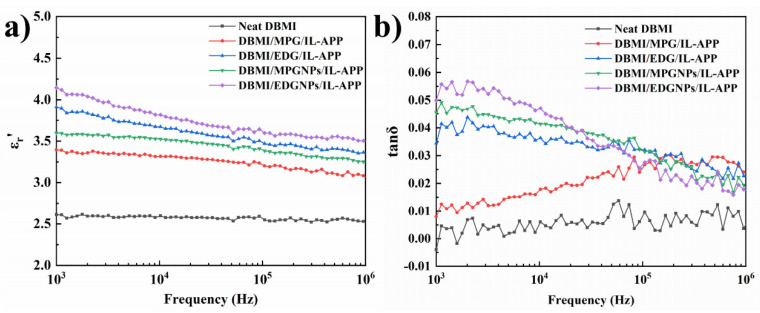
Relationships between dielectric constant (**a**) and frequency and between dielectric loss tangent (**b**) and frequency of Neat DBMI, DBMI/MPG/IL-APP, DBMI/EDG/IL-APP, DBMI/MPGNPs/IL-APP, DBMI/EDGNPs/IL-APP.

**Table 1 materials-14-06406-t001:** Formulae of Neat DBMI and DBMI composites.

Sample	DABA (wt.%)	BMI (wt.%)	MPG (wt.%)	EDG (wt.%)	MPGNPs ^a)^ (wt.%)	EDGNPs ^b)^ (wt.%)	IL (wt.%)	APP (wt.%)
Neat DBMI	44.44	55.56	--	--	--	--	--	--
DBMI/MPG/IL-APP	40.00	50.00	0.50	--	--	--	0.50	9.00
DBMI/EDG/IL-APP	40.00	50.00	--	0.50-	--	--	0.50	9.00
DBMI/MPGNPs/IL-APP	40.00	50.00	--	--	1.00	--	--	9.00
DBMI/EDGNPs/IL-APP	40.00	50.00	--	--	--	1.00	--	9.00

Notes ^a)^ &^b)^: MPGNPs and EDGNPs contain 1.13 wt.% and 1.36 wt.% GNPs, respectively.

**Table 2 materials-14-06406-t002:** The micro scale of MPGNPs and EDGNPs in selected *X*-axis segments.

Sample	Horizontal Distance (μm)	Vertical Distance (nm)	Surface Distance (μm)
MPGNPs	0.1	7.6	0.1
EDGNPs	0.1	2.7	0.1

**Table 3 materials-14-06406-t003:** TGA test data for Neat DBMI and DBMI composites.

Sample	T_d,5%_ (°C)	T_max_ (°C)	WLR_max_ (% min^−1^)	R_450 °C_ (%)	R_700 °C_ (%)
Neat DBMI	409.3 ± 0.2	453.1 ± 0.2	0.99	67.9 ± 0.2	27.2 ± 0.2
DBMI/MPG/IL-APP	374.0 ± 0.1	422.5 ± 0.2	0.39	72.6 ± 0.2	54.6 ± 0.2
DBMI/EDG/IL-APP	375.3 ± 0.2	422.4 ± 0.2	0.32	75.9 ± 0.2	59.2 ± 0.2
DBMI/MPGNPs/IL-APP	376.2 ± 0.2	418.3 ± 0.2	0.38	74.1 ± 0.2	59.2 ± 0.2
DBMI/EDGNPs/IL-APP	369.8 ± 0.2	415.2 ± 0.2	0.33	74.6 ± 0.2	59.9 ± 0.2

**Table 4 materials-14-06406-t004:** CONE test results of Neat DBMI and DBMI composites.

Sample	TTI (S)	PHRR (kW·m^−2^)	THR (MJ·m^−2^)	TSR (m^2^·m^−2^)	av-EHC (MJ·kg^−1^)	av-MLR (g·s^−1^)
Neat DBMI	40 ± 3	327.1 ± 4.5	83.8 ± 0.2	2616.1 ± 0.2	22.8 ± 0.2	0.06 ± 0.007
DBMI/MPG/IL-APP	37 ± 3	239.7 ± 14.2	54.2 ± 0.5	1571.3 ± 0.5	22.3 ± 0.5	0.04 ± 0.003
DBMI/EDG/IL-APP	43 ± 3	291.2 ± 36.1	57.9 ± 0.5	1492.4 ± 0.5	21.7 ± 0.5	0.04 ± 0.002
DBMI/MPGNPs/IL-APP	42 ± 3	184.4 ± 10.8	54.4 ± 0.4	1357.6 ± 0.4	21.5 ± 0.4	0.04 ± 0.003
DBMI/EDGNPs/IL-APP	39 ± 3	190.4 ± 17.0	43.7 ± 0.3	1248.0 ± 0.2	22.5 ± 0.2	0.03 ± 0.003

**Table 5 materials-14-06406-t005:** Mass lost percentage and LOI of DBMI composites after water resistance (120 h) tests.

Sample	Mass Lost (wt.%)	LOI-after Soaking (%)	△LOI (%)
DBMI/APP	2.36	35.3	5.6
DBMI/IL-APP	2.09	35.9	5.3
DBMI/MPG/IL-APP	1.71	36.7	5.2
DBMI/EDG/IL-APP	1.40	37.2	4.4
DBMI/MPGNPs/IL-APP	1.26	40.6	3.2
DBMI/EDGNPs/IL-APP	0.96	40.9	2.8

## Data Availability

The data presented in this study are available in article.

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
