# Peer review of "Graphene Nanoplatelets Hybrid Flame Retardant Containing Ionic Liquid and Ammonium Polyphosphate for Modified Bismaleimide Resin: Excellent Flame Retardancy, Thermal Stability, Water Resistance and Unique Dielectric Properties"

_materials, 2021, doi:10.3390/ma14216406_

Round 1

Reviewer 1 Report

Authors report on polymer based nanocomposites for flame retardant applications, these are my suggestions to improve the manuscript:
1.    I would add the chemical formula when introducing BMI
2.    Please reformulate the abstract in present tense, avoid acronyms, avoid too technical details while stressing the main quantitative or qualitative result
3.    English style improvement and proofreading by a professional agency could be highly beneficial to increase the impact of the work
4.    Section 3: please revise reasonable digits according to error bars, for, for example: “7.6 nm and the surface radius was about 0.1 μm. The aspect ratio is about 26, the thickness of GNPs in EDGNPs is 2.6 nm, and surface radius is about 0.1 μm”. When giving quantitative evolution of performances there is no discussion on statistics, error bars, reproducibility, and the precision of evaluations/comparisons looks to me overestimated for most of the cases.
5.    L209: “…but both of them have lower defect density” lower than what?
6.    L214-25 please summarize data without reporting them in the text
7.    The text is full of acronym that makes it very specialized to a technical audience, I suggest to better explain acronyms to reach a broader/less technical audience as the community of Materials. For example authors could introduce more clearly LOI and CONE tests.

Reviewer 2 Report

The presented text was focused on the development of new type of DBMI resin fire retardant. In particular research authors were evaluating the effectiveness of ammonium polyphosphate APP flame retardant modified with ionic liquid and graphene. The scientific value of the work is, in my opinion, at a very high level. The concept of the conducted work, research methodology and diligence in presenting the results indicate the high knowledge of the authors. Some minor comments presented below might improve the paper quality.

  1. I am not convinced about the nano scale of graphene additives. AFM images were prepared using scans of the MPGNP or EDGNP surface, which does not constitute clear evidence of the presence of exfoliated structures in final composite. My assumptions are confirmed by the C element scan maps, which indicate the agglomeration of the structure.
  2. Due to the fact that the flammability of composites is the most important part of the work, it would be worth including the appearance of samples after LOI or cone calorimeter tests.
